# Should I stay or should I go? A retrospective propensity score-matched analysis using administrative data of hospital-at-home for older people in Scotland

Apostolos Tsiachristas,[1] Graham Ellis,[2] Scott Buchanan,[3] Peter Langhorne,[4] David J Stott,[4] Sasha Shepperd[5]

For numbered affiliations see end of article.

**Correspondence to**
Dr Apostolos Tsiachristas;
apostolos.tsiachristas@ndph.ox.ac.uk

## ABSTRACT

**Objectives** To compare the characteristics of populations admitted to hospital-at-home services with the population admitted to hospital and assess the association of these services with healthcare costs and mortality.

**Design** In a retrospective observational cohort study of linked patient level data, we used propensity score matching in combination with regression analysis.

**Participants** Patients aged 65 years and older admitted to hospital-at-home or hospital.

**Interventions** Three geriatrician-led admission avoidance hospital-at-home services in Scotland.

**Outcome measures** Healthcare costs and mortality.

**Results** Patients in hospital-at-home were older and more socioeconomically disadvantaged, had higher rates of previous hospitalisation and there was a greater proportion of women and people with several chronic conditions compared with the population admitted to hospital. The cost of providing hospital-at-home varied between the three sites from £628 to £2928 per admission. Hospital-at-home was associated with 18% lower costs during the follow-up period in site 1 (ratio of means 0.82; 95% CI: 0.76 to 0.89). Limiting the analysis to costs during the 6 months following index discharge, patients in the hospital-at-home cohorts had 27% higher costs (ratio of means 1.27; 95% CI: 1.14 to 1.41) in site 1, 9% (ratio of means 1.09; 95% CI: 0.95 to 1.24) in site 2 and 70% in site 3 (ratio of means 1.70; 95% CI: 1.40 to 2.07) compared with patients in the control cohorts. Admission to hospital-at-home was associated with an increased risk of death during the follow-up period in all three sites (1.09, 95% CI: 1.00 to 1.19 site 1; 1.29, 95% CI: 1.15 to 1.44 site 2; 1.27, 95% CI: 1.06 to 1.54 site 3).

**Conclusions** Our findings indicate that in these three cohorts, the populations admitted to hospital-at-home and hospital differ. We cannot rule out the risk of residual confounding, as our analysis relied on an administrative data set and we lacked data on disease severity and type of hospitalised care received in the control cohorts.

## INTRODUCTION

Organising health systems to optimise the health outcomes of older people and contain costs is a priority as populations around the world age, and the demand for healthcare continues to rise. Despite a global policy emphasis on 'care closer to home'[1] and initiatives that seek to ease demand for hospital based healthcare, efforts to innovate and deliver healthcare services that provide an alternative to hospital admission for older people have been piecemeal and often lack a health system perspective. A lack of evidence to support decision-making has contributed to this. Avoiding admission to hospital by providing acute healthcare in people's homes, often as a hospital outreach service, is one of the more popular service innovations and yet there is uncertainty around the effectiveness and cost-effectiveness of this form of care.[2]

The use of administrative data to evaluate service delivery interventions has the potential to provide a simple and efficient mechanism to provide real-world evidence about policy-relevant service innovations, and embed evaluation into local decision-making. However, previous experience of using routine data in this area of research has been of mixed success due to a limited set of

variables, missing data and the complexity of policy-relevant questions that often require a broad and longer term perspective.[3] Administrative healthcare data collected in Scotland is unique in that it is population based, with little missing data. The aim of this study was to use these data to compare the characteristics of populations from three Health Boards who used a geriatrician-led hospital-at-home service with the population who received hospital care, and to assess the impact of these services on healthcare costs and mortality.

## METHODS

### Setting

We used patient level data collected by three of the fourteen Scottish Health Boards of all patients aged 64 years and older, and who were admitted (referred to as the index admission) to either geriatrician-led admission avoidance hospital-at-home or inpatient hospital between August 2014 and December 2015 (17 months) in site 1 and site 2, and between January 2015 and December 2016 (24 months) in site 3. These services are commissioned by integrated health and social care boards that cover a population of almost 1.5 million in urban and rural areas. The Information Service Division (ISD), part of National Health Service Scotland, deidentified, cleaned and linked individual patient records to derive activity and costs related to periods before and after the index admissions. We obtained signed release forms from each Board's Caldicott guardian, and followed the ISD data sharing agreement.

### Intervention

The three service models of hospital-at-home provided an admission avoidance function that provided an alternative to inpatient hospital care, and had similar structures and functions; the main differences were in the capacity of the services and the organisation of services for rehabilitation (box 1).

### Data sources

Data were available for each person for 2 years prior to their index admission, and from the point of their index admission to 6 months after index discharge from hospital-at-home or hospital. Box 2 presents a full list of all variables included in the dataset. Figure 1 provides schematic

---

**Box 1    Description of each service**

**Hospital-at-home**

The three hospital-at-home services are broadly similar, capacity ranged between 24 and 60 beds for the period of the analysis. Each is a geriatrician-led service that is supported by nurses (sometimes nurse practitioners) and therapy practitioners for the initial assessment; geriatricians and the multidisciplinary team review patients in their homes and meet daily (a virtual ward round) to discuss patient cases and agree actions. Rehabilitation is available within the existing team with onward referral to community rehabilitation as required, and in one site rehabilitation is accessed through a parallel community rehabilitation services. Out of hours emergency cover is provided by primary care out-of-hours. Patients are referred to the service from GPs, sometimes through a central referral number or via step down from the acute hospital. The service offers access to diagnostics such as radiology, and intravenous fluids, antibiotics and oxygen. Cases are discussed daily with the multidisciplinary team at the virtual ward round and daily management plans agreed. In one site, there is close working with the day hospital where patients can be referred for follow-up or for investigations. Patients access investigations and treatment with the same speed as inpatients. The services support intravenous therapies in the home.

**Hospital**

The provision of hospital based acute health services varied among the sites; in one site, there were three district general hospitals (1653 beds) that provide acute health services to a mainly urban population of 652 230, with a total of 1653 beds; in site 2 a hospital (550 beds) provides acute healthcare to a population of 180 130; and in site 3 there are two district general hospitals (825 beds) that provide healthcare to a population of 358 900, and acute admissions are via one of the hospitals.

---

**Box 2    List of variables included in the dataset**

Costs of accidents and emergency attendances,
Costs of acute day cases,
Costs of acute elective hospitalisation,
Costs of acute non-elective hospitalisation,
Costs of geriatric wards,
Costs of mental health wards,
Costs of outpatient visits,
Costs of prescribed medication,
Costs of (re)admission to hospital-at-home.
Primary ICD-10 codes on index discharge,
Secondary ICD-10 codes on index discharge,
Length of stay of the index admission,
Age on index admission,
Gender,
Scottish Index of Multiple Deprivation, 1 (most deprived) to 10 (most affluent),
Long-term conditions,
Date of death (if applicable).
Based on ICD-10 codes:
Cardiovascular disease (I60-I69, G45),
Chronic obstructive pulmonary disorder (J41-J44, J47),
Dementia (F00-F03, F05.1),
Diabetes (E10-E14),
Coronary heart disease (ICD10: I20-I25),
Heart failure (I500, I501, I509),
Renal failure (N03, N18, N19, I12, I13),
Epilepsy (G40, G41),
Asthma (J45, J46),
Atrial fibrillation (I48, MS, G35),
Cancer (C00-C97),
Arthritis (M05, M19, M45, M47, M460-M462, M464, M468, M469),
Parkinson's (G20-G22),
Chronic liver disease (K711, K713, K714, K717, K754),
Congenital problems (Q00-Q99),
Diseases of blood and blood forming organs (D50-D89),
Other diseases of the digestive system (K00-K122, K130-K839, K85X, K860-K93),
Other endocrine metabolic diseases (E00-E07, E15-E35, E70-E90),
Admitted to hospital-at-home or hospital.

---

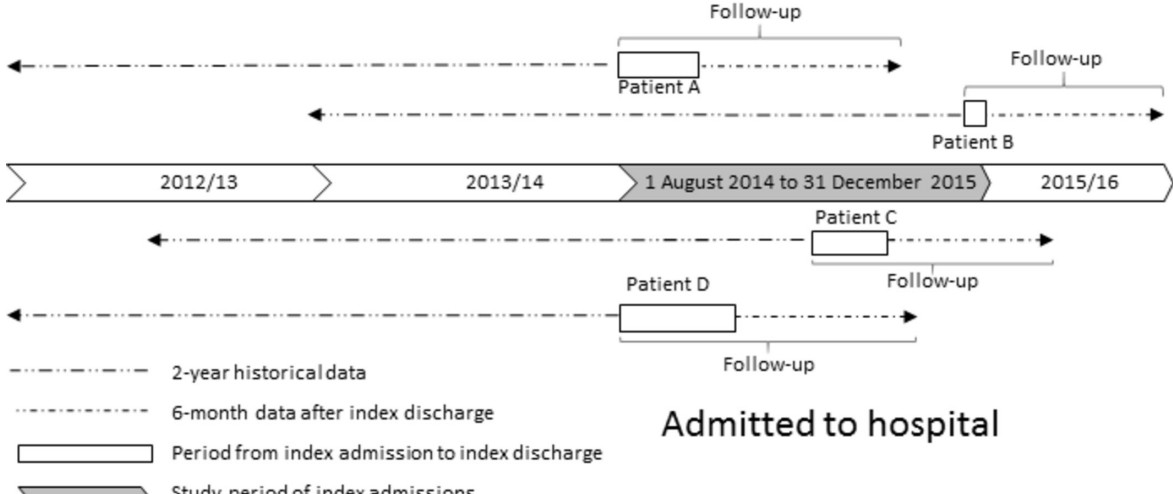

**Figure 1**  Illustration of obtained data from site 1.

examples of the differing calendar time periods studied before and after index admission for people admitted between August 2014 and December 2015 to hospital-at-home (patients A and B) or hospital (patients C and D) in site 1. As this illustrates, the maximum follow-up period for each patient consisted of the period between index admission and index discharge and 6 months after index discharge. The data were collected via the data systems used in hospitals to collect patient data. Hospital-at-home activity data was submitted to ISD from the local systems of the three sites. The linked data set included acute inpatient, geriatric long stay and day case, mental health admissions, outpatient appointments accident and emergency attendances, community prescribing and death registrations.

### Selection of patients in the hospital-at-home and control cohorts

We included patients aged 65 years and older, and who were classified as an unscheduled admission to general or geriatric medicine. In the control cohort, we excluded those with a diagnosis that would not be eligible for management through hospital-at-home; these exclusions included acute intracerebral crisis (intracerebral infections, trauma or haemorrhage), stroke and related codes, acute coronary syndromes and myocardial infarction, surgical emergencies including vascular, urological, gynaecological and general surgical presentations, orthopaedic diagnosis of fractures and trauma, cardiothoracic diagnoses, poisoning and complications of surgery. We also excluded from the control group those who had a diagnosis (ie, primary and secondary International Classification of Diseases (ICD-10) code) that was not observed in any of the hospital-at-home admissions in each site (1081 patients in site 1, 1405 in site 2 and in 451 in site 3) (figure 2). Each patient was counted as a single episode of healthcare.

### Intervention costs

We collected data on the costs of hospital-at-home using a template derived from the Cost-It tool of the WHO.[4] The cost categories included staff, training, transport, information and communication, clinical materials/equipment, support services, laboratory services, diagnostics, overheads and other costs. Clinician managers supported by finance staff in the three Health Boards completed this template based on the actual spending for the hospital-at-home service for the time periods covered by the ISD data. The cost per hospital-at-home admission was calculated by dividing the total costs of the hospital-at-home service by the total number of hospital-at-home admissions during the same period.

### Statistical analysis

We used an iterative approach to the analysis, starting with a description of the two cohorts (ie, those admitted to hospital-at-home and those admitted to hospital) for each Health Board. We calculated means, SD and frequencies to describe differences in patient characteristics at index admission and tested differences using two sample t-test and Mann-Whitney test for continues variables and $\chi^2$ test for categorical variables. We also estimated the mean differences in resource utilisation costs (with bootstrapped SEs) and the unadjusted relative risk of mortality between the two cohorts for each Health Board.

Further, we investigated the association of being admitted to hospital-at-home or hospital with mortality and cost over a minimum follow-up period of 6 months. To do this, we followed the Medical Research Council guidelines on performing natural experiments and scientific literature to adopt a stepwise strategy to select the propensity score matching (PSM) technique that most reduced observed confounding between the two cohorts in each Health Board.[5–8] First, we included all possible confounding variables available in the dataset (see box 2

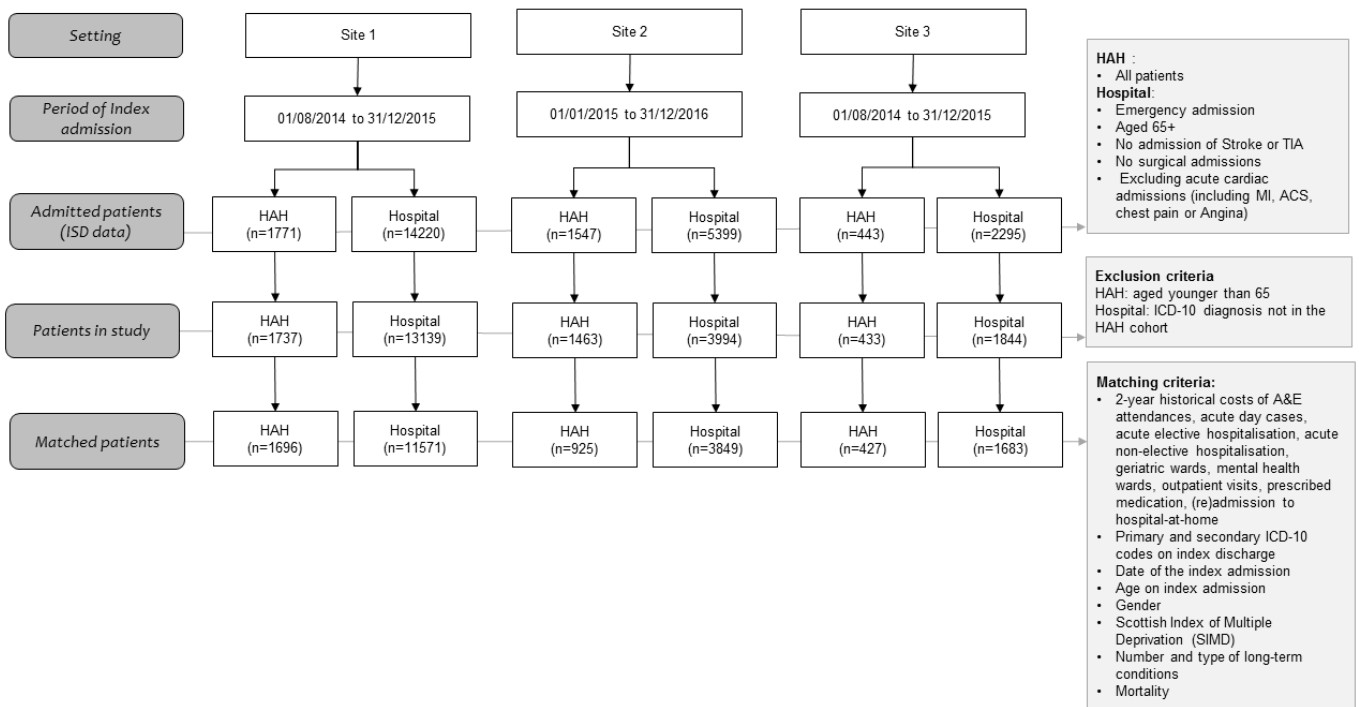

**Figure 2** Flow chart of study population. ISD, Information Service Division.

and figure 2), and considered that the inclusion of covariates not associated with the treatment assignment would have little influence in the propensity score model.[5] Second, we matched the two cohorts in each site using a range of the most commonly used PSM techniques; these included Mahalanobis, 1-to-1, K-to-1, kernel, local linear regression, spline and inverse probability weighting techniques. Second, the performance of each PSM technique on covariate balancing was assessed based on the mean and median percentage standardised bias as well as Rubin's B (the absolute standardised difference of the means of the linear index of the propensity score in the treated and (matched) non-treated group) and Rubin's R (the ratio of treated to (matched) non-treated variances of the propensity score index). Following Rubin's (2001) recommendation, we considered B less than 25 and R between 0.5 and 2 to indicate sufficient balance.[9] Third, we chose the PSM technique that had the lowest values on these performance indicators in each of the three Health Boards. We matched the two cohorts in each Health Board by sociodemographic characteristics (ie, age, gender, socioeconomic status), diagnosis code (ie, primary and secondary ICD-10 code) of index admission, morbidity (ie, type of long-term condition, mortality during follow-up (for the analysis of cost), 2-year costs prior to the index admission (by cost category as listed in box 1), and date of index admission (to account for seasonal trends).

We performed a doubly robust estimation to further reduce confounding by using a regression analysis after performing the most suitable PSM technique and

including the confounding variables listed above as covariates.[10] In the regression, we used generalised linear regression models (GLMs) with gamma distribution and log link to investigate the association of hospital-at-home with total costs during the follow-up period, and total costs in 6 months following index discharge. We also used GLMs with Poisson distribution and log link to estimate the relative risk of mortality. Robust SEs were specified in all regression models. We calculated Kaplan-Meier survival curves, with and without using the weights from the PSM, and used log-rank tests to test the equality of the survival functions. There were few missing observations in the dataset and thus, complete case analysis was performed.

### Subgroup analysis
We conducted a sub-group analysis, running the same regression models used in the main analysis, to investigate the association of hospital-at-home services with costs and mortality for the population who had a diagnosis of dementia. We considered this population to be important due to their complex healthcare needs, and the increasing prevalence of dementia.[11 12] In a second subgroup analysis, we excluded patients who died during the follow-up period and investigated the association of hospital-at-home with costs. In both subgroup analyses, PSM was performed to match sub-cohorts in each site.

### Sensitivity analysis
In a univariate sensitivity analysis, we reduced and increased the intervention cost of admission avoidance

hospital-at-home by 50%, as there are no standard unit costs to benchmark these types of services and we were concerned that costs for these services may vary due to economies of scale, size, experience, setting, human resource capacity and error. This sensitivity analysis was expected to impact the costs during index admission and the costs of admission to hospital-at-home in the 6 months after discharge. In another sensitivity analysis, we estimated the E-value to assess how strong unmeasured confounding would have to be with both the treatment (ie, admission to hospital-at-home) and outcome (ie, costs and mortality) to fully explain away the estimated treatment effects, conditional on the measured confounders.[13 14]

### Patient involvement
Patients were not involved in this retrospective analysis of administrative data.

## RESULTS
### Characteristics of the population cohorts
After applying the exclusion criteria, 1737 patients were admitted to hospital-at-home in site 1 between August 2014 and December 2015 (17 months), 1463 patients were admitted to hospital-at-home in site 2 between January 2015 and December 2016 (24 months), and 433 patients were admitted to hospital-at-home in site 3 between August 2014 and December 2015 (17 months) (figure 2). In the same period, there were 13 139 patients admitted to three hospitals in site 1, 3994 patients admitted to one hospital in site 2, and 1844 patients admitted to one hospital in site 3.

There were few differences between the hospital-at-home cohorts in the three sites, the main difference being that a larger proportion of the population in site 2 lived in a more affluent area (ie, scored five or higher on the Scottish Index of Multiple Deprivation). Patients admitted to hospital-at-home were on average three to 4 years older than those admitted to hospital, were more likely to be female (range from five percentage points to nine percentage points), and a higher proportion had more than four long-term conditions (approximately seven percentage points) compared with patients admitted to hospital (table 1). The largest difference between those admitted to hospital-at-home and to hospital in site 1 and site 2 was in the proportion of patients with dementia (10 percentage points higher in the hospital-at-home cohorts), while in site 3, it was the proportion of patients with renal failure (also 10 percentage points higher in the hospital-at-home cohort).

We compared the two cohorts in each site, from index admission to 6 months postdischarge from hospital-at-home or hospital (table 2). There was on average a higher percentage of deaths while receiving healthcare in hospital compared with those receiving healthcare in hospital-at-home (6% vs 1%, site 1; 6% vs 3%, site 2; 4% vs 1%, site 3); and a higher percentage of deaths in

the follow-up period, from admission to 6 months after discharge, in the groups that had received hospital-at-home (21% vs 28%, site 1; 22% vs 32%, site 2; 17% vs 27%, site 3). Patients in the hospital-at-home cohort lived on average 8 (site 1), 10 (site 2) and 12 (site 3) fewer days during the whole follow-up, and their index admission was on average fewer days in site 1 (mean unadjusted difference −2.64, 95% CI −2.97 to −2.31) and site 3 (mean unadjusted difference −2.02, 95% CI −2.66 to −1.37) and longer in site 2 (mean unadjusted difference 1.25, 95% CI 0.86 to 1.64).

The cost during a hospital-at-home admission was on average lower than hospital admission in site 1 (mean difference −£2318; 95% CI: £−2420 to £−2217) and site 3 (mean difference −£1096; 95% CI: −£1398 to −£793), and slightly lower (mean difference −£153; 95% CI: −£277 to −£29) in site 2 (table 2). In the hospital-at-home cohort, these costs included the intervention costs of delivering the service at home, which were £628 per admission and £113 per day in site 1, £2928 per admission and £398 per day in site 2 and £864.54 per admission and £117.57 per day in site 3. In each Health Board, staff were the major driver of the cost of delivering hospital-at-home (site 1 95%, site 2 87%, site 3 94%). Detailed information on the costs of delivering hospital at home are in online supplementary appendix 1.

Each of the three hospital-at-home cohorts incurred higher healthcare costs, driven by non-elective hospitalisation, prior to their index admission compared with the respective control cohort. Site 1 had on average 40% higher costs (mean difference £3219; 95% CI: £2513 to £3925), site 2 56% higher costs (mean difference £5064; 95% CI: £3984 to £6143) and site 3 57% higher costs (mean difference £4115; 95% CI: £2467 to £5764). In the 6 months following discharge from the index admission, costs were higher for each of the three hospital-at-home cohorts; in site 1 costs were on average 43% higher (mean difference £1839; 95% CI: £1423 to £2255), in site 2, they were 16% higher (mean difference £875, 95% CI: £156 to £1595), and in site 3, they were 92% higher (mean difference £3068, 95% CI: £2178 to £3958). The larger increase in costs in all sites was due to higher non-elective hospitalisation costs in the group who had received hospital-at-home care (mean difference £1517, 95% CI £1134 to £1899 site 1; mean difference £529, 95% CI −£77 to −£1135 site 2; mean difference £2618, 95% CI £1779 to £3458 site 3) during the 6 months follow-up.

When the cost of the index admission was included in the analysis, the cost during follow-up (ie, including the index admission and 6 months healthcare resource use after index discharge) was 6% lower (mean difference -£480, 95% CI: £−996 to £36) in the hospital-at-home cohort, compared with the control cohort in site 1; while these costs were 8% higher in site 2 (mean difference £722, 95% CI: £32 to £1413) and 35% higher in site 3 (mean difference £1973, 95% CI: £1019 to £2927).

Compared with the control cohort, the mean costs per day of being alive during the follow-up period were 13%

**Table 1** Patient characteristics at index admission

| Variable | Site 1 Control (n=13 139) | Site 1 HAH (n=1737) | Site 2 Control (n=3994) | Site 2 HAH (n=1463) | Site 3 Control (n=1844) | Site 3 HAH (n=433) |
|---|---|---|---|---|---|---|
| Mean age on admission (SD) | 77.8 (7.78) | 81.2 (7.21)** | 78.5 (8.11) | 82.2 (7.82)** | 77.3 (7.81) | 81.4 (7.12)** |
| Female | 7468 (57%) | 1096 (63%)** | 2102 (53%) | 909 (62%)** | 1037 (56%) | 266 (61%)* |
| Higher than four on the SIMD | 5005 (38%) | 609 (35%)** | 1960 (49%) | 775 (53%)* | 837 (45%) | 192 (44%) |
| More than four chronic conditions | 4974 (38%) | 777 (45%)** | 1664 (42%) | 725 (50%)** | 659 (36%) | 185 (43%)** |
| Arthritis | 3431 (26%) | 497 (29%)* | 1455 (37%) | 572 (39%) | 606 (33%) | 155 (36%) |
| Asthma | 1370 (10%) | 183 (11%) | 497 (13%) | 207 (14%) | 177 (10%) | 49 (11%) |
| Atrial fibrillation | 3659 (28%) | 488 (28%) | 1555 (29%) | 468 (32%)* | 498 (27%) | 126 (29%) |
| Cancer | 3749 (29%) | 485 (28%) | 1261 (32%) | 371 (25%)** | 580 (31%) | 124 (29%) |
| CVD | 2922 (22%) | 467 (27%)** | 763 (19%) | 392 (27%)** | 373 (20%) | 114 (26%)** |
| Liver disease | 499 (4%) | 50 (3%) | 183 (5%) | 52 (4%) | 72 (4%) | 20 (5%) |
| COPD | 3641 (28%) | 505 (29%) | 1083 (27%) | 428 (29%) | 510 (28%) | 132 (31%) |
| Dementia | 1999 (15%) | 439 (25%)** | 665 (17%) | 390 (27%)** | 223 (12%) | 74 (17%)** |
| Diabetes | 2985 (23%) | 403 (23%) | 948 (24%) | 350 (24%) | 410 (22%) | 115 (27%)* |
| Epilepsy | 459 (4%) | 75 (4%) | 146 (4%) | 78 (5%)** | 53 (3%) | 10 (2%) |
| CHD | 5034 (38%) | 733 (42%)** | 1425 (36%) | 575 (39%)* | 624 (34%) | 141 (33%) |
| Heart failure | 2197 (17%) | 404 (23%)** | 744 (19%) | 32 (23%)** | 328 (18%) | 109 (25%)** |
| MS | 73 (1%) | 6 (0%) | 21 (1%) | 17 (1%)* | 14 (1%) | 2 (1%) |
| Parkinson's | 293 (2%) | 66 (4%)** | 82 (2%) | 53 (4%)** | 53 (3%) | 20 (5%) |
| Renal failure | 2501 (19%) | 394 (23%)** | 780 (20%) | 339 (23%)** | 284 (15%) | 110 (25%)** |
| Congenital problems | 277 (2%) | 38 (2%) | 159 (4%) | 51 (4%) | 51 (3%) | 9 (2%) |
| Diseases of blood | 3784 (29%) | 553 (32%)** | 1143 (29%) | 426 (29%) | 485 (26%) | 125 (29%) |
| Endocrine metabolic disease | 4505 (34%) | 624 (36%) | 1737 (44%) | 652 (45%) | 642 (35%) | 151 (35%) |
| Disease of digestive system | 9341 (71%) | 1249 (72%) | 2710 (68%) | 1006 (69%) | 1145 (62%) | 286 (66%) |

A patient could be registered with more than one ICD-10 codes.
SIMD ranges from 1 (most deprived) to 10 (most affluent).
*P<0.05; **P<0.01 in $\chi^2$ test for categorical and two sample t-test and Mann-Whitney test for continuous variables to test differences between HAH and control.
CHD, coronary heart disease COPD, chronic obstructive pulmonary disorder; CVD, cardiovascular disease; HAH, hospital-at-home; MS, multiple sclerosis; SIMD, Scottish Index of Multiple Deprivation.

(mean difference −£12; 95% CI: −£17 to −£6) lower in the hospital-at-home cohort in site 1, while these costs were 34% higher (mean difference £37; 95% CI: £18 to £56) and 66% higher (mean difference £36; 95% CI: £18 to £53) in site 2 and site 3, respectively.

### Selection of PSM technique
In the propensity score matched analysis, there were 1696, 925 and 427 patients in the hospital-at-home cohort and 11571, 3849 and 1683 patients in the hospital cohort in site 1, site 2 and site 3, respectively (figure 2). Local linear regression matching was the best PSM technique to match the cohorts in site 1 and site 3 for costs and mortality, as it resulted in a lower mean (ie, 1.5 and 1.8 respectively) and median (ie, 1.2 and 1.6 respectively) percentage

standardised bias, as well as the lowest Rubin's B (ie, 9.4 and 9.6 respectively). Based on the same criteria, Kernell matching was selected to match the cohorts in site 2. Rubin's R was within the suggested range (ie, from 0.5 to 2) in the selected techniques. These results as well as the patient characteristics at index admission after PSM are presented in online supplementary appendix 2. As this appendix shows, the differences in patient characteristics between the compared cohorts were almost eliminated after PSM.

### Main propensity score matched analysis
The results of the main analysis are presented in panel A in table 3. After PSM and regression analysis, the healthcare cost for site 1 in hospital-at-home during the whole

**Table 2** Mortality, resource utilisation and costs

| Variable | Site 1 | | | Site 2 | | | Site 3 | | |
|---|---|---|---|---|---|---|---|---|---|
| | Control (n=13 139) | HAH (n=1737) | Mean difference or risk ratio (95% CI) | Control (n=3994) | HAH (n=1463) | Mean difference or risk ratio (95% CI) | Control (n=1844) | HAH (n=433) | Mean difference or risk ratio (95% CI) |
| Died during index admission | 844 (6%) | 20 (1%) | 0.18 (0.12 to 0.28)* | 256 (6%) | 47 (3%) | 0.50 (0.37 to 0.68)* | 78 (4%) | 2 (1%) | 0.11 (0.03 to 0.44)* |
| Died during follow-up including index admission | 2787 (21%) | 483 (28%) | 1.31 (1.21 to 1.42)* | 867 (22%) | 471 (32%) | 1.48 (1.35 to 1.63)* | 319 (17%) | 116 (27%) | 1.55 (1.29 to 1.86)* |
| Mean days alive during follow-up (SD) | 159 (57) | 151 (60) | −8.32 (−11.32 to −5.32) | 156 (57) | 146 (66) | −10.10 (−14 to −7) | 163 (52) | 151 (60) | −12 (−18 to −6) |
| Mean length of index admission in days (SD) | 8.18 (13.13) | 5.54 (5.23) | −2.64 (−2.97 to −2.31) | 6.10 (8.74) | 7.35 (5.50) | 1.25 (0.86 to 1.64) | 6.36 (11.27) | 4.34 (4.19) | −2.02 (−2.66 to −1.37) |
| Mean 2-year historical costs (SD) | | | | | | | | | |
| Accident and emergency | 173 (260) | 253 (289) | 80 (65 to 94) | 136 (224) | 180 (238) | 44 (28 to 60) | 143 (214) | 202 (248) | 59 (31 to 87) |
| Elective hospital care | 985 (4183) | 956 (5586) | −28 (−352 to 295) | 1027 (4040) | 705 (3287) | −321 (−519 to −123) | 981 (3733) | 1036 (7738) | 55 (−723 to 833) |
| Non-elective hospital care | 4037 (9051) | 6945 (11078) | 2908 (2452 to 3364) | 5101 (11716) | 9593 (15081) | 4492 (3804 to 5179) | 3978 (9063) | 7832 (12784) | 3854 (2591 to 5118) |
| Hospital day case | 707 (2868) | 439 (1318) | −269 (−340 to −197) | 625 (4186) | 290 (1676) | −336 (−479 to −193) | 544 (2121) | 358 (1139) | −186 (−334 to −38) |
| Geriatric long stay | 360 (3078) | 504 (3430) | 143 (−66 to 354) | 117 (1824) | 252 (2757) | 135 (−13 to 283) | 105 (1321) | 229 (1221) | 125 (14 to 235) |
| Mental ward | 247 (3637) | 367 (4865) | 119 (−177 to 411) | 347 (5019) | 1053 (7839) | 706 (265 to 1147) | 220 (3231) | 252 (2903) | 32 (−329 to 393) |
| Outpatient | 173 (204) | 173 (200) | 0 (−11 to 11) | 222 (244) | 206 (232) | −15 (−30 to 0) | 212 (270) | 201 (253) | −11 (−38 to 15) |
| Medication (GP prescriptions) | 1468 (1675) | 1733 (1796) | 256 (187 to 341) | 1524 (1738) | 1883 (1989) | 360 (253 to 466) | 1034 (1661) | 1221 (1621) | 188 (30 to 346) |
| Total | 8149 (12538) | 11369 (14951) | 3219 (2513 to 3925) | 9098 (239) | 14162 (477) | 5064 (3984 to 6143) | 7217 (11478) | 11333 (16071) | 4115 (2467 to 5764) |
| Mean costs during index admission (SD) | 3195 (4683) | 877† (1336) | −2318 (−2420 to −2217) | 3426 (4473) | 3273† (1217) | −153 (−277 to −29) | 2383 (3872) | 1287 (2753) | −1096 (−1398 to −793) |
| Mean costs 6 months after index discharge (SD) | | | | | | | | | |
| Accident and emergency | 72 (130) | 88 (117) | 17 (11 to 22) | 55 (124) | 53 (105) | −2 (−9 to 4) | 59 (101) | 71 (113) | 12 (−1 to 25) |
| Elective hospital care | 305 (2284) | 157 (1642) | −148 (−236 to −60) | 272 (1781) | 204 (1928) | −68 (−190 to 53) | 169 (1433) | 313 (2440) | 144 (−92 to 380) |
| Non-elective hospital care | 2444 (5885) | 3961 (7124) | 1517 (1134 to 1899) | 3942 (8203) | 4471 (9597) | 529 (−77 to 1135) | 2029 (5281) | 4648 (8767) | 2618 (1779 to 3458) |
| Hospital day case | 237 (1230) | 73 (440) | −164 (−191 to −138) | 234 (1485) | 96 (804) | −139 (−198 to −79) | 168 (985) | 63 (320) | −105 (−162 to −48) |
| Geriatric long stay | 643 (5191) | 1014 (5467) | 371 (79 to 663) | 218 (2158) | 150 (1753) | −68 (−178 to 41) | 320 (2400) | 700 (3873) | 381 (−73 to 834) |
| Mental ward | 165 (2539) | 206 (2113) | 41 (−58 to 140) | 299 (3508) | 259 (2928) | −40 (−224 to 143) | 211 (2803) | 120 (1291) | −91 (−245 to 64) |
| Outpatient | 54 (108) | 45 (95) | −9 (−13 to −5) | 61 (116) | 54 (105) | −8 (−14 to −2) | 65 (128) | 67 (131) | 2 (−12 to 16) |

Continued

**Table 2** Continued

| Variable | Site 1 | | | Site 2 | | | Site 3 | | |
|---|---|---|---|---|---|---|---|---|---|
| | Control (n=13 139) | HAH (n=1737) | Mean difference or risk ratio (95% CI) | Control (n=3994) | HAH (n=1463) | Mean difference or risk ratio (95% CI) | Control (n=1844) | HAH (n=433) | Mean difference or risk ratio (95% CI) |
| Medication (GP prescriptions) | 392 (515) | 415 (540) | 23 (−5 to 52) | 402 (546) | 482 (627) | 80 (45 to 115) | 314 (504) | 338 (566) | 24 (−28 to 76) |
| Hospital-at-home | 4 (56) | 196 (446) | 193 (170 to 216) | 50 (444) | 642 (1737) | 592 (506 to 679) | 7 (59) | 90 (257) | 83 (59 to 108) |
| Total | 4316 (8928) | 6155 (9990) | 1839 (1423 to 2255) | 5535 (9734) | 6410 (10919) | 875 (156 to 1595) | 3342 (6990) | 6410 (10614) | 3068 (2178 to 3958) |
| Mean costs in follow-up (SD) including index admission | 7513 (10510) | 7031 (10110) | −480 (−996 to 36) | 8961 (11394) | 9683 (11072) | 722 (32 to 1413) | 5724 (8523) | 7697 (10834) | 1973 (1019 to 2927) |
| Mean costs per lived day in follow-up (SD) | 83 (150) | 72 (114) | −12 (−17 to −6) | 109 (178) | 146 (304) | 37 (18 to 56) | 55 (96) | 91 (165) | 36 (18 to 53) |

*Unadjusted risk ratio.
†It includes the interventions costs (ie, £628 in site 1, £2928 in site 2 and £865.54 in site 3) and other costs occurred during the episode.
GP, general practice; HAH, hospital-at-home.

follow-up period (ie, during index admission and over 6 months after discharge from the index admission) was on average 18% lower (ratio of means: 0.82; 95% CI: 0.76 to 0.89) than admission to hospital. When the cost of the index admission was excluded from the hospital-at-home and hospital cohorts, costs were on average 27% higher (ratio of means: 1.27; 95% CI: 1.14 to 1.41) for hospital-at-home compared with hospital in site 1. In site 2, the difference in costs between the hospital-at-home and hospital was close to zero (ratio of means: 1.00; 95% CI 0.92 to 1.09) during the whole follow-up period and 9% higher (although not statistically significant) (ratio of means: 1.09; 95% CI: 0.95 to 1.24) when the cost of the index admission was excluded. In site 3, patients admitted to hospital-at-home had on average 15% higher (although not statistically significant) cost during the whole follow-up period (ratio of means: 1.15; 95% CI 0.99 to 1.33) and 70% higher cost when the cost of the index admission was excluded (ratio of means: 1.70; 95% CI 1.40 to 2.07) compared with patients admitted to hospital. The full results of the regression analyses are presented in online supplementary appendix 3.

There may be an increased risk of mortality in all three hospital-at-home cohorts (site 1: relative risk 1.09; 95% CI 1.00 to 1.19) (site 2: relative risk 1.29; 95% CI: 1.15 to 1.44) (site 3: relative risk 1.27; 95% CI: 1.06 to 1.54) compared with the hospital cohort after PSM and regression were performed to adjust for confounding. The Kaplan-Meier survival curves presented in figure 3 show higher survival rates in the inpatient control cohorts in all three sites, and after weighting with the propensity score the control cohort in site 2 continued to have a higher survival rate than the hospital-at-home cohort. The difference in survival in site 3 between the results reported in table 3 and the survival curve after weighting is explained by the fact that Kaplan-Meier curves are only weighted with the propensity score without performing an additional regression analysis.

### Results of the subgroup analysis

Patients with dementia (Panel B in table 3) admitted to hospital-at-home services in site 1 and site 2 had an average of 24% lower costs (site 1: ratio of means 0.76; 95% CI 0.66 to 0.87; site 2: ratio of means 0.76 95% CI: 0.66 to 0.88) from the index admission to 6 months post-discharge. We found that the population who were admitted to hospital-at-home, and had a diagnosis of dementia, may have an increased risk of death (site 1: 1.05, 95% CI 0.89 to 1.24; site 2: relative risk 1.41, 95% CI 1.19 to 1.67; site 3: relative risk 1.65, 95% CI 1.12 to 2.41) compared with those who had a diagnosis of dementia and who were admitted to hospital.

When we excluded people who died during follow-up (ie, during index admission and 6 months after discharge), patients admitted to hospital-at-home in site 1 had lower costs (ratio of means 0.85, 95% CI: 0.77 to 0.94), while there was 11% increase in costs in site 2 (ratio of means 1.11, 95% CI: 1.00 to 1.25) and 20%

**Table 3** Results of the propensity score matched regression analyses

| Panel A: main analysis | | | |
|---|---|---|---|
| *Outcome variable* | *Site 1 (n=13 267)* | *Site 2 (n=4769)* | *Site 3 (n=2110)* |
| Total costs during follow-up period* | 0.82 (0.03) [0.76 to 0.89]<0.001 | 1.00 (0.05) [0.92 to 1.09] 0.982 | 1.15 (0.09) [0.99 to 1.33] 0.073 |
| Total costs in 6 months after discharge | 1.27 (0.07) [1.14 to 1.41]<0.001 | 1.09 (0.07) [0.95 to 1.24] 0.219 | 1.70 (0.17) [1.40 to 2.07]<0.001 |
| Mortality rate during follow-up | 1.09 (0.05) [1.00 to 1.19] 0.059 | 1.29 (0.07) [1.15 to 1.44]<0.0010 | 1.27 (0.12) [1.06 to 1.54] 0.011 |
| **Panel B: subgroup analysis including only patients with dementia** | | | |
| *Outcome variable* | *Site 1 (n=2321)* | *Site 2 (n=1053)* | *Site 3 (n=280)* |
| Total costs during follow-up period* | 0.76 (0.05) [0.66 to 0.87]<0.001 | 0.76 (0.06) [0.66 to 0.88]<0.001 | 0.87 (0.15) [0.63 to 1.21] 0.409 |
| Total costs in 6 months after discharge | 1.18 (0.11) [0.99 to 1.41] 0.071 | 0.75 (0.09) [0.59 to 0.96] 0.021 | 1.58 (0.41) [0.95 to 2.63] 0.078 |
| Mortality rate during follow-up | 1.05 (0.09) [0.89 to 1.24] 0.594 | 1.41 (0.12) [1.19 to 1.67]<0.001 | 1.65 (0.32) [1.12 to 2.41] 0.011 |
| **Panel C: subgroup analysis including only survivors** | | | |
| *Outcome variable* | *Site 1 (n=10 132)* | *Site 2 (n=3584)* | *Site 3 (n=1691)* |
| Total costs during follow-up period* | 0.85 (0.04) [0.77 to 0.94] 0.002 | 1.11 (0.03) [1.00 to 1.25] 0.058 | 1.20 (0.11) [1.00 to 1.43] 0.046 |
| Total costs in 6 months after discharge | 1.23 (0.08) [1.08 to 1.40] 0.002 | 1.17 (0.10) [0.99 to 1.38] 0.070 | 1.71 (0.20) [1.36 to 2.15]<0.001 |
| **Panel D: sensitivity analysis** | | | |
| *Outcome variable* | *Site 1 (n=13 267)* | *Site 2 (n=4769)* | *Site 3 (n=2110)* |
| Total costs during follow-up period* (assuming 50% lower intervention costs) | 0.77 (0.03) [0.71 to 0.84]<0.001 | 0.81 (0.04) [0.74 to 0.9] 0.001 | 1.07 (0.09) [0.91 to 1.25] 0.399 |
| Total costs during follow-up period* (assuming 50% higher intervention costs) | 0.87 (0.03) [0.81 to 0.94] 0.001 | 1.18 (0.05) [1.09 to 1.28]<0.001 | 1.23 (0.09) [1.07 to 1.42] 0.004 |

The results are presented as coefficient (SE) (95% CI) p value. The results are after matching and adjusting for age, gender, socioeconomic status, primary and secondary ICD-10 codes of index admission, type of long-term condition, mortality (for the analysis of costs), 2-year costs prior to the index admission (by cost category as listed in box 1).
 *It includes the index admission period and 6 months postdischarge.

increase in site 3 (ratio of means 1.20, 95% CI: 1.00 to 1.43); the mean costs were higher in the hospital-at-home cohort when the costs during the index admission were excluded (site 1: ratio of means 1.23, 95% CI: 1.08 to 1.40; site 2: ratio of means 1.17, 95% CI 0.99 to 1.38; site 3: ratio of means 1.71, 95% CI 1.36 to 2.15) compared with patients admitted to hospital (Panel C in table 3).

## Results of the sensitivity analyses

The results from the sensitivity analysis (Panel D in table 3) showed that patients in the hospital-at-home cohort in site 1 had 13% lower costs (ratio of means 0.87; 95% CI: 0.81 to 0.94) during the follow-up period (ie, during index admission and 6 months after index discharge) when the hospital-at-home service costs were assumed to be 50% higher than in the main analysis. In site 2, the results from the sensitivity analysis showed that the uncertainty in hospital-at-home service costs lead to increased costs or cost savings by about 18% (ratio of means 1.18; 95% CI: 1.09 to 1.28) during the whole follow-up period. In site 3, the sensitivity analysis showed a 23% cost increase (ratio of means 1.23; 95% CI: 1.07 to 1.42), if the intervention costs of hospital-at-home were 50% higher. The estimated e-values are presented in online supplementary appendix 4 and show that unmeasured confounders should be strongly associated with admission to hospital-at-home as well as with costs and mortality after adjusting for the observed confounders in order to explain away the results of the main analysis.

## DISCUSSION

### Main findings

Patients who received healthcare from the hospital-at-home services were older, were more socioeconomically

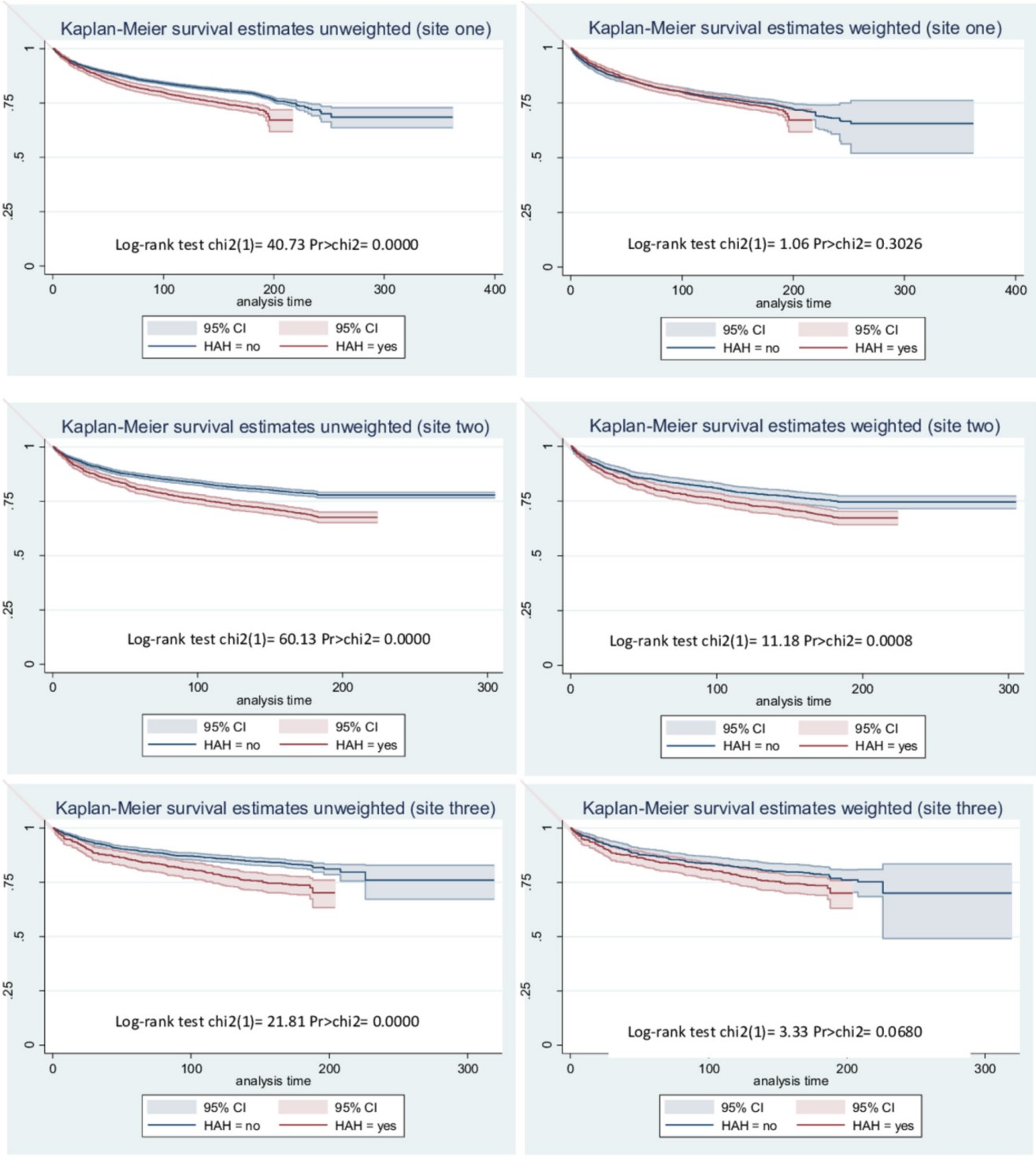

Note: The cohorts in each site were matched on age, gender, socio-economic status, primary and secondary ICD-10 codes of index admission, type of long-term condition, 2-year costs prior to the index admission (by cost category as listed in Box 1); Weighted refers to weighting the observation of each patient based on the propensity score to be in the hospital-at-home cohort as described in the propensity score matching section.

**Figure 3**  Survival curves before and after propensity score matching. HAH, hospital-at-home.

disadvantaged, had higher morbidity (measured by the number of long term conditions), higher rates of previous hospitalisation and there was a greater proportion of women compared with the group admitted to hospital. The two groups also differed in terms of their clinical diagnosis, with the most marked difference across the three services being a greater percentage (5%–10% difference) of people with dementia. The higher healthcare costs over the 2 years prior to index admission in those admitted to hospital-at-home were mainly driven by the

costs of non-elective hospitalisation. However, the differences in patient characteristics were almost eliminated after PSM. The cost of providing hospital-at-home varied between the three sites from £628 to £2928 per admission, and costs were driven primarily by staff costs. Our findings indicate that hospital-at-home might be associated with an increase in healthcare costs in the 6 months after index discharge. However, this increase in costs might be offset by likely cost-savings during the index admission. The higher healthcare cost at 6 months after index discharge, was driven primarily by acute non-elective hospitalisation. Interpreting this is not straightforward; it might indicate a lack of resources during the index admission to hospital-at-home, or an increased risk of hospital admission in the population who receive their healthcare through hospital-at-home. The suggestion of an increased risk of mortality at 6 months after the index admission might be genuine, or could indicate that PSM did not control for all differences between the groups and thus, the estimates are subject to residual confounding.[15 16]

## Comparison with previous studies

A meta-analysis of six small randomised controlled trials concluded that admission avoidance hospital-at-home probably makes little or no difference to the risk of death or transfer to hospital at 6 months' follow-up, and might increase the likelihood of living at home (although with low-certainty evidence); and highlighted the lack of evidence on cost.[2] Studies that have used 'real life data' offer the potential to address criticisms of limited external validity from randomised trials; and PSM is one technique that has been used to balance co-variates when analysing routinely collected health data to assess these type of service delivery interventions. Findings have been consistent, and previous studies have reported higher rates of mortality and unplanned admission for those who received an intermediate care intervention, compared with matched controls.[6 16 17] However, it is possible that these findings are subject to residual confounding.

## Potential mechanisms and interpretation

Healthcare services that cross the interface of primary and secondary care can bridge and strengthen the integration of acute and community services, and social care. However, by definition this can lead to a complex arrangement of services that reflect availability of local resources,[18] and a willingness to innovate. The hospital-at-home services evaluated in this analysis were established to reduce the demand for acute hospital beds by providing an alternative to admission to hospital, and to lower the risk of functional decline from the limited mobility that older people might experience when in hospital. However, it is possible that the services have several functions, for example by providing both rapid response and reablement, and this is reflected in the diverse population included in this analysis. Existing services and the overall structure of the healthcare care system in Scotland may also have influenced the shape and scope of hospital-at-home functions.

Regarding the control cohorts, older people admitted to acute hospital in Scotland receive quite variable care and access to comprehensive geriatric assessment depending on whether they are placed in a geriatric medical unit or other environments such as general adult medicine. This variation may also have influenced the results of this study.

## Implications for clinicians and policy-makers

The variation in intervention costs of the three hospital-at-home services is primarily driven by staff costs, and the findings of the sensitivity analysis confirms that staff costs are likely to determine whether a hospital-at-home service leads to higher costs or cost savings. The skill-mix of healthcare professionals who provide hospital-at-home should be guided by national standards, the type of patients the service targets and the function of the service in terms of whether or not the service supplements existing community based healthcare, substitutes for hospital level care, augments palliative care services or a combination of these. The integration of these types of service with existing primary and secondary care services, for example the provision of out-of-hours care by primary care services, might also determine the costs of these services. Managerial capacity of these services is expected to be of crucial importance in setting-up and managing the team of professionals able to provide high quality care.

The absence of evidence based guidelines about who and under which conditions a patient may be admitted to admission avoidance hospital-at-home might explain the variation in the set-up of services, the difference in patient characteristics between patients admitted to hospital-at-home and hospital, and the relatively small size of the services. This is confirmed by the National Audit of Intermediate Care,[19] that was established in response to concerns about governance structures in intermediate care services, and reported a complex pattern of service provision.

Data on the role and capability of informal care givers is largely absent. In many cases, people admitted to hospital-at-home services receive care from their partners who if old might have health issues themselves.

## Strengths and limitations

The strengths of this study include the dataset from three of the largest Health Boards in Scotland, the quasi-experimental study design that has allowed inferences from real world evidence, and the sensitivity analyses that helped to address uncertainty in the results. The major limitation of this type of non-randomised comparison is residual confounding. While matching individuals and performing regression analysis can reduce this risk, it is possible that the two populations differed in frailty because we did not match and adjust for differences in the use of community and social services prior to index admission. If unobserved confounders were part of the clinical-decision making by GPs and geriatricians to admit patients to

hospital-at-home or hospital, our findings might be biased due to confounding by clinical indication. This type of confounding is often not measured directly because standardised criteria are not available to guide clinical decision-making.[20 21] Therefore, the magnitude of this bias in our results depends on the clinical-decision making process to admit patients to hospital-at-home in the three sites. If clinicians did not consider hospital-at-home as a substitute service to hospitalisation then confounding by indication would increase the residual confounding in our analysis. GPs and geriatricians who refer patients to hospital-at-home are likely to have a clinical bias in preferring to keep older, frailer and terminally ill patients in their own home. Using hospital-at-home admission criteria to define the control cohort accepts that such open criteria will include general medical patients who are likely to have fewer comorbidities, be younger and with a longer life expectancy. However, as the results of the survivors' subgroup analysis were very similar with the results of the main cost analysis we expect that the magnitude of the residual confounding to be small. Furthermore, the use of routine data has been used to reliably identify older people with frailty,[22] and approaches using clinical codes to define this population are being tested.[23]

## Future research
Guidance on the use of real life data to evaluate service delivery interventions is largely absent, and could provide healthcare decision-makers with a relatively inexpensive way of evaluating local service innovations and how to avoid pitfalls in analysis and interpretations. Similar to all observational studies, the findings of this study may be used to identify important questions to be tested in randomised trials.[20] A multicentre randomised trial that measures outcomes that are key to decision-makers (including informal care giving), and is accompanied by a process evaluation to help explain the findings, is necessary to provide clinicians and policy-makers with further evidence about the effectiveness and cost-effectiveness of admission avoidance hospital-at-home services across UK. The authors are involved in such a trial the results of which are expected to be available in 2019.[24]

## CONCLUSIONS
We found differences in the populations admitted to hospital-at-home and hospital. The likely higher cost in all three hospital-at-home cohorts, compared with the hospital cohorts during the 6 months following discharge, highlights the importance of characterising populations eligible to receive these types of healthcare services and of assessing subsequent use of health, social and informal care following admission to hospital-at-home or hospital. The lack of data on the severity of the observed acute and chronic conditions as well as on type of hospitalised care received in the control cohorts means that we cannot rule out the risk of residual confounding, and the findings should be interpreted with caution.

**Author affiliations**
¹Health Economics Research Centre, Nuffield Department of Population Health, University of Oxford, Oxford, UK
²Monklands Hospital, NHS Lanarkshire, Airdrie, UK
³Information Services Division, National Services Scotland, Edinburgh, UK
⁴Institute of Cardiovascular and Medical Sciences, University of Glasgow, Glasgow, UK
⁵Nuffield Department of Population Health, University of Oxford, Oxford, UK

**Acknowledgements** We would like to thank Charmaine Walker, Jenny Boyd, Alistair Smith and Josh Matthews from ISD Scotland for providing us with the data as well as Christine McGregor (economist in the Scottish Government) for her insightful views and expertise. We are also indebted to Dr Mike Gardner and Prof Alastair Gray (both University of Oxford), Prof Stavros Petrou (University of Warwick), and Dr Matthew Sperrin (University of Manchester) for commenting on previous drafts of the manuscript. Our thanks also to Prof Gillian Parker (University of York), Dr Angela Coulter (University of Oxford) and Prof Stuart Parker (University of Newcastle) for their useful reflection on the study findings. We would also like to thank all healthcare staff in all three sites who made this study happen. AT acknowledges support by the NIHR Oxford Biomedical Research Centre.

**Contributors** AT, GE and SS were responsible for study concept. GE and SB facilitated the acquisition of data. AT and SS led the writing of the protocol, study design and drafting of the manuscript. AT performed the statistical analysis. PL and DJS provided clinical expertise and commented on previous versions of the manuscript. All authors interpreted the data, critically revised the manuscript for important intellectual content and approved the final version for submission. AT and SS are guarantors.

**Funding** NIHR, UK. (12/5003//01; "How to Implement Cost-Effective Comprehensive Geriatric Assessment").

**Competing interests** GE is leading one of the hospital-at-home services in this study.

**Patient consent for publication** Not required.

**Ethics approval** We obtained local data transfer agreements and signed release forms from each Health Board's Caldicott guardian. Further approval from an ethics committee was not required because the study was part of a service audit and the data provided to the researchers were deidentified.

**Provenance and peer review** Not commissioned; externally peer reviewed.

**Data sharing statement** No additional data are available.

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
