## [Reviewer comments · BMJ Open]

ARTICLE DETAILS

TITLE (PROVISIONAL)	Should I stay or should I go? A retrospective propensity score matched analysis using administrative data of hospital-at-home for older people in Scotland
AUTHORS	Tsiachristas, Apostolos; Ellis, Graham; Buchanan, Scott; Langhorne, Peter; Stott, David; Shepperd, S

VERSION 1 - REVIEW

REVIEWER	Nicolas Roche University Paris Descartes, France
REVIEW RETURNED	11-Jun-2018

GENERAL COMMENTS	This matched retrospective observational study suggests that geriatric hospital-at-home has deleterious effects both on costs and survival. This is an important result that should lead to caution and efforts to improve the quality of hospital at home for these patients. The statistics appear adequate. However, as outlined by the authors, ruling out residual confounding in this type of study is impossible. An important factor in this regard is the lack of medical data on the severity of observed acute and chronic diseases. Thus it is impossible to conclude anything firmly from this study. However this paper has the merit of outlining the uncertainty and the need for further studies. Demographic and socio-economic factors seem to play an important role in the decision to refer patients to hospital at home, which should be discussed more. The influence of the healthcare system on the results could also be more extensively discussed.
--

REVIEWER	Christian Backer Mogensen University of Southern Denmark Denmark
REVIEW RETURNED	16-Sep-2018

GENERAL COMMENTS	Thank you for getting the chance to review this manuscript. After having tried several times to review the manuscript, I have to admit my limitations in time, statistical and health economic knowledge. The paper is 48 pages long, easy to read and follow. The background is short and relevant and opens the research question adequately. The aim is very relevant. I cannot assess the statistical methodology adequately, I recommend a statistician and a health economist to review instead. discussion: balanced, easy to follow, with fine discussion of limitation. If the results after proper review by statisticians and economists are valid, the importance of these results is very relevant for the ongoing discussion of the hospital at home concept: it is more expensive and associated with higher mortality. My main worry is, that the manuscript in its present form is too comprehensive and technically difficult for a clinical researcher to read.
--

REVIEWER	Elizabeth Buckley University of South Australia
REVIEW RETURNED	22-Oct-2018

GENERAL COMMENTS	I have only reviewed the statistical analysis, methods and strengths and limitations of the manuscript. The investigators have done well to account confounding to the extent that they can given the setting (observational and real world evidence) and administrative data available. They have attempted to account for confounding between cohorts through propensity matching. As the authors indicate, the greatest concern with propensity matching of the cohorts is the potential for residual confounding which cannot be ruled out particularly when using administrative (and at times limited) data. However, through subgroup analysis, they have indicated that the results are robust and residual confounding is likely to be small. The caveats around the methodology used, and results, could be reiterated in the Abstract and conclusion.
--

VERSION 1 – AUTHOR RESPONSE

Reviewer 1

Comment: Demographic and socio-economic factors seem to play an important role in the decision to refer patients to hospital at home, which should be discussed more.

Authors' reply: We discuss the observed differences in patient characteristics between those admitted to hospital and those admitted to hospital-at-home in a sentence in the Discussion that reads:

“The absence of evidence based guidelines about who and under which conditions a patient may be admitted to admission avoidance hospital-at-home might explain the variation in the set-up of services, the difference in patient characteristics between patients admitted to hospital-at-home and hospital, and the relatively small size of the services”.

We also made clear that these differences were almost eliminated after propensity score matching. This is shown in a new table in Appendix 2 in the Supplementary File. We refer to this new table in the “Selection of propensity score matching technique” in a revised sentence that reads: “These results as well as the patient characteristics at index admission after propensity score matching are presented in Appendix 2. As this Appendix shows, the differences in patient characteristics between the compared cohorts were almost eliminated after propensity score matching.” We have also added a sentence in the first paragraph of the Discussion that reads “However, the differences in patient characteristics were almost eliminated after propensity score matching.”

Comment: The influence of the healthcare system on the results could also be more extensively discussed.

Authors’ reply: We elaborated on the role of the healthcare system in a revised ending of the “Potential mechanisms and interpretation” section of the Discussion that reads “However, it is possible that the services have several functions, for example by providing both rapid response and reablement, and this is reflected in the diverse population included in this analysis. Existing services and the overall structure of the healthcare care system in Scotland may also have influenced the shape and scope of hospital-at-home functions. Regarding the control cohorts, older people admitted to acute hospital in Scotland receive quite variable care and access to comprehensive geriatric assessment depending on whether they are placed in a geriatric medical unit or other environments such as general adult medicine. This variation may also have influenced the results of this study.”

Reviewer 2

Comment: My main worry is, that the manuscript in its present form is too comprehensive and technically difficult for a clinical researcher to read.

Authors’ reply: This is indeed an observational study in which statistical analysis was performed to deal with observed confounding. We have revised the manuscript and edited to make it more accessible.

Reviewer 3

Comment: The caveats around the methodology used, and results, could be reiterated in the Abstract and conclusion.

Authors’ reply: We have added text to reiterate the limitations of the analysis in the conclusion of the Abstract and the Discussion of the paper. In the conclusions section in the Abstract this now reads “Our findings indicate that in these three cohorts, the populations admitted to hospital-at-home and hospital differ. We cannot rule out the risk of residual confounding, as our analysis relied on an administrative data set and we lacked data on the severity of disease.” and added a last sentence in the Conclusions that reads “The lack of data on the severity of the observed acute and chronic conditions as well as on type of hospitalised care received in the control cohorts means that we cannot rule out the risk of residual confounding, and the findings should be interpreted with caution.”

VERSION 2 – REVIEW

REVIEWER	Yevgeniya Gokun University of Arkansas for Medical Sciences
REVIEW RETURNED	12-Feb-2019

GENERAL COMMENTS	1. Table 1: I am curious why you presented standard error for mean age on admission. Usually mean along with SD is presented. Also you used Mann-Whitney test for continuous variables to test the differences between HAH and control. If non-parametric testing is used, mean and IQR is presented. Have you performed two sample t-tests to see if the p-values match with your non-parametric approach? 2. Section (Main Propensity score matched analysis): you mention several times that the cost is either 9% or 15% higher when your 95% CI includes 1 therefore it is not statistically significant result. You can mention it is higher but add that's it was not statistically significant. Perhaps for those instances, including p-values in the text would be beneficial just to see how far it is from 0.05. 3. You performed sensitivity analysis by subsetting your intervention cost of admission. Have you performed another one to see how well your PSM did? As you aware, PSM included measures confounders but I am curious would you get the same results if you were to determine the sensitivity of the observed effect of your treatment on outcome to unmeasured confounders?
---

VERSION 2 – AUTHOR RESPONSE

Reviewer 4

Comment: Table 1: I am curious why you presented standard error for mean age on admission. Usually mean along with SD is presented. Also you used Mann-Whitney test for continuous variables to test the differences between HAH and control. If non-parametric testing is used, mean and IQR is presented. Have you performed two sample t-tests to see if the p-values match with your non-parametric approach?

Authors' reply: In the revised manuscript, the standard deviations of the means are provided in Table 1 and Table 2 and the second sentence in the Statistical Analysis section now reads "We calculated means, standard deviations, and frequencies to describe differences in patient characteristics at index admission and tested differences using two sample t-test and Mann-Whitney test for continuous variables and Chi-square test for categorical variables."

Comment: Section (Main Propensity score matched analysis): you mention several times that the cost is either 9% or 15% higher when your 95% CI includes 1 therefore it is not statistically significant result. You can mention it is higher but add that's it was not statistically significant. Perhaps for those instances, including p-values in the text would be beneficial just to see how far it is from 0.05.

Authors' reply: We follow the reviewer's suggestion to help the readers to interpret the 95%CI in the results and we added "(although not statistically significant)" in these two instances.

Comment: You performed sensitivity analysis by subsetting your intervention cost of admission. Have you performed another one to see how well your PSM did? As you aware, PSM included measures confounders but I am curious would you get the same results if you were to determine the sensitivity of the observed effect of your treatment on outcome to unmeasured confounders?

Authors' reply: Thank you for this useful suggestion. We performed a bias analysis/sensitivity analysis to assess the impact of unmeasured confounding on our results. With regards to this analysis, we added the following text in the Sensitivity Analysis section: "In another sensitivity analysis, we estimated the E-value to assess how strong unmeasured confounding would have to be with both the treatment (i.e. admission to hospital-at-home) and outcome (i.e. costs and mortality) to fully explain away the estimated treatment effects, conditional on the measured confounders.^{13 14}" as well as in Results of the Sensitivity Analyses section "The estimated E-Values are presented in Appendix 4 and

show that unmeasured confounders should be strongly associated with admission to hospital-at-home as well as with costs and mortality after adjusting for the observed confounders in order to explain away the results of the main analysis.” We also provided the estimated E-values and graphs in Appendix 4 in the Supplementary File.